# Tailoring exciton and excimer emission in an exfoliated ultrathin 2D metal-organic framework

Wei-Ming Liao[1], Jian-Hua Zhang[1], Shao-Yun Yin[1], He Lin[2], Xingmin Zhang[2], Jihong Wang[2], Hai-Ping Wang[1], Kai Wu[1], Zheng Wang[1], Ya-Nan Fan[1], Mei Pan[1] & Cheng-Yong Su[1]

Two-dimensional (2D) metal–organic frameworks have exhibited a range of fascinating attributes, of interest to numerous fields. Here, a calcium-based metal-organic framework with a 2D layered structure has been designed. Dual emissions relating to intralayer excimers and interlayer trapped excitons are produced, showing excitation-dependent shifting tendency, characteristic of a low dimensional semiconductor nature. Furthermore, the layer stacking by weak van der Waals forces among dynamically coordinated DMF molecules enables exfoliation and morphology transformation, which can be achieved by ultrasound in different ratios of DMF/$H_2O$ solvents, or grinding under appropriate humidity conditions, leading to nano samples including ultrathin nanosheets with single or few coordination layers. The cutting down of layer numbers engenders suppression of interlayer exciton-related emission, resulting in modulation of the overall emitting color and optical memory states. This provides a rare prototypical model with switchable dual-channel emissions based on 2D-MOFs, in which the interlayer excitation channel can be reversibly tuned on/off by top-down exfoliation and morphology transformation.

[1] MOE Laboratory of Bioinorganic and Synthetic Chemistry, State Key Laboratory of Optoelectronic Materials and Technologies, Lehn Institute of Functional Materials, School of Chemistry, Sun Yat-Sen University, 510275 Guangzhou, China. [2] Shanghai Synchrotron Radiation Facility, Shanghai Institute of Applied Physics, Chinese Academy of Sciences, 201204 Shanghai, China. Correspondence and requests for materials should be addressed to M.P. (email: panm@mail.sysu.edu.cn)

Metal-organic frameworks (MOFs) have been investigated for application in a variety of fields including adsorption, catalysis, sensing, drug delivery, and so on[1–5]. Besides composition and structure, shape, dimension and morphology also play important roles in determining the physical and chemical properties of MOFs, and special performances have been found in micro-structured and/or nano-structured MOFs which can not be observed in bulk solids. For this purpose, various synthetic methods, such as solvothermal, electrochemical, microwave or ultrasound, etc, have been developed to modify the dimension and morphology of MOFs[6]. However, at this stage, the design and syntheses of expected micro/nano-MOFs with specific morphologies and related properties still remains more of an art than a science[7].

2D-MOFs, as a representative type of low dimensional materials[8–15], have attracted special interest in recent years, due to their unique attributes exhibited in such fields as gas adsorption and separation[16–20], biomimetic enzyme[21], sensing and detection[22–25], catalysis[26–28], chemical release, etc[29–34]. Especially, in view of light and optics, 2D nano-structures might have pronounced dimensional confinement effect to change the energy states and modulate the excitation and emission properties in MOFs[35]. To obtain appropriate 2D-MOFs, apart from the bottom-up assembly strategies starting from initial metal and organic building units, facile post-synthetic top-down approaches, such as liquid exfoliation under ultrasound conditions, have been used in transferring infinitely layered MOFs into 2D nanosheets comprising several sheets or even one sheet of metal-organic coordination structures by breaking down the weak interactions between layers[36–38]. This constitutes a promising method to obtain low dimensional MOFs while maintaining the intralayer metal-ligand coordination environment as that in bulk crystals. In addition, mechanical exfoliation can also be applied to obtain MOF nanosheets, similar to the situation in obtaining single-layer graphene from graphite through a cleavage method[39]. For example, grinding with mortar or milling balls has been proven to be an efficient strategy to obtain MOF nanosheets from bulk crystals[40–42]. Typically, these morphological transformations happen via a solid-solid or solid-liquid interface, and the latter usually involves the mediation of solvents. Comparatively, exfoliation via a solid-gas interface has been barely explored so far. Only few reported works can be found, for example, to exfoliate boron nitride into nanosheets by liquid-nitrogen gasification[43].

Herein, a 2D calcium metal-organic framework (Ca–MOF, LIFM-41, $[Ca_3(HL)_2(DMF)_5]_n$) (DMF = N, N-dimethylformamide, $H_4L$ = 2'-amino-[1,1':4',1"-terphenyl]-3,3",5,5"-tetracarboxylic acid) with van der Waals layered crystal structure, was synthesized via a solvothermal reaction between $H_4L$ and $CaCl_2$. The 2D-MOF can be further exfoliated and morphologically modified by ultrasound or grinding procedure. Due to the reversible formation/breakage of multi-layered coordination structure during morphological transformations, the interlayer excitation pathway can be tuned on/off to manipulate the overall emitting color and optical memory states for the 2D-MOF, which prompts a new concept for the fabrication of MOF-based optoelectronic devices.

## Results

**X-ray crystal structure of 2D Ca-MOF**. The calcium 2D-MOF, $[Ca_3(HL)_2(DMF)_5]_n$ (Ca-MOF, LIFM-41) was synthesized by solvothermal reaction of $H_4L$ and $CaCl_2$ in $DMF/H_2O$ (v/v = 5/2) solution, and characterized by single-crystal and powder X-ray diffraction (SCXRD and PXRD), Fourier transform infrared (FT-IR) spectroscopy, and thermogravimetric (TG) analyses (see supporting information for details, Supplementary Fig. 1 and Supplementary Table 1). From the crystal structure shown in Fig. 1, we can see that Ca–MOF crystallizes in C2/c space group, and the asymmetric unit consists of 1.5 $Ca^{2+}$ ions, one $HL^{3-}$ ligand, and 2.5 coordinated DMF molecules. In the coordination environment of metal centers, there are two kinds of $Ca^{2+}$ ions. Ca1 is octa-coordinated by eight oxygen atoms, among which five are from three different $HL^{3-}$ ligands, and the other three are from DMF molecules. While Ca2 is hexa-coordinated with carboxylate O atoms from six different $HL^{3-}$ ligands, showing slightly distorted octahedral geometry. Each $HL^{3-}$ ligand acts as a $\mu_6$-bridge to link six $Ca^{2+}$ ions using three of its four carboxylate groups, in which two carboxylate groups adopt a $\mu_2$-$\eta^2$:$\eta^1$-bridging mode, and one adopts a $\mu_2$-$\eta^1$:$\eta^1$-bridging coordination mode. While the fourth carboxylate group remains non-ligated to $Ca^{2+}$ ion. Besides the non-coordinated carboxylate group, there is also a vacant –$NH_2$ group on each $HL^{3-}$ ligand in the crystal structure of Ca–MOF, which contributes not only to its long-wavelength absorption property, but also provides the possibility of further post-modification on this site.

In general, in the Ca-MOF crystal structure, {$Ca_3O_{18}$} metal-oxygen coordinated cluster-centers are formed. Through the linkage between $Ca^{2+}$ ions and $HL^{3-}$ ligands based on these clusters, Z-like metal-ligand chains are formed (Fig. 1e), and further extended into 2D layers in Ca-MOF. It is interesting to notice that the formed 2D layers are pendent with coordinated DMF molecules, and packing together via linkage of weak van der Waals interactions among DMF (Fig. 1f and Supplementary Fig. 2). The coordination of DMF molecules are labile in nature, and therefore, the 2D layered structure in Ca-MOF can be delaminated post-synthetically by breaking the bonds and weak interactions among them. This important character provides the potential to regulate the morphology of Ca-MOF by ultrasound or grinding exfoliation, which will be shown below.

**Photophysical properties of calcium 2D-MOF bulk crystals**. For comparison, the UV-vis absorption spectrum of $H_4L$ ligand was first measured in DMF solvent (Supplementary Fig. 3). The peak below 320 nm should be ascribed to E band of aromatic groups, and the broad peak centered at 350 nm was attributed to K band of aromatic groups with a red shift resulting from auxochrome amine group. Emission spectra of $H_4L$ ligand in solid and DMF solvent were measured and compared (Supplementary Fig. 4). In solid state, there are mainly two emission peaks, for which the higher energy one at 390 nm was ascribed to interligand charge transfer (ILCT) emission, while the lower energy emission at 550 nm was attributed to excimers associated with direct coupling of the ligand molecules' excited and ground states. To further validate this assignment, the emission properties of $H_4L$ were studied in DMF solvent. In high concentration, both the ILCT and excimer emission peaks appeared, but with dozens-nm of blue-shifts compared with the solid state (shifting to 340 and 515 nm, respectively). While in low concentration DMF solution, the low energy emission peak longer than 500 nm would disappear, manifesting its excimer-related attributes. It is noted that both the positions of ILCT and excimer emissions of $H_4L$ ligand do not shift with the changing of excitation wavelengths. This is different from the excitation-dependent dual-emissions in Ca-MOF bulk crystals, which will be discussed in detail below.

After coordination, the as-prepared Ca-MOF bulk crystal samples in sub-millimeter scale showed a strong absorption band at around 330 nm with a shoulder peak extending above 500 nm (Supplementary Fig. 5). Excitation spectra monitored at various emission wavelengths were measured for Ca-MOF, which exhibit two main peaks at 408 and 477 nm with varied intensities at different monitoring wavelengths (Fig. 2a). While the emission spectra of Ca-MOF bulk crystals display uncommon excitation-

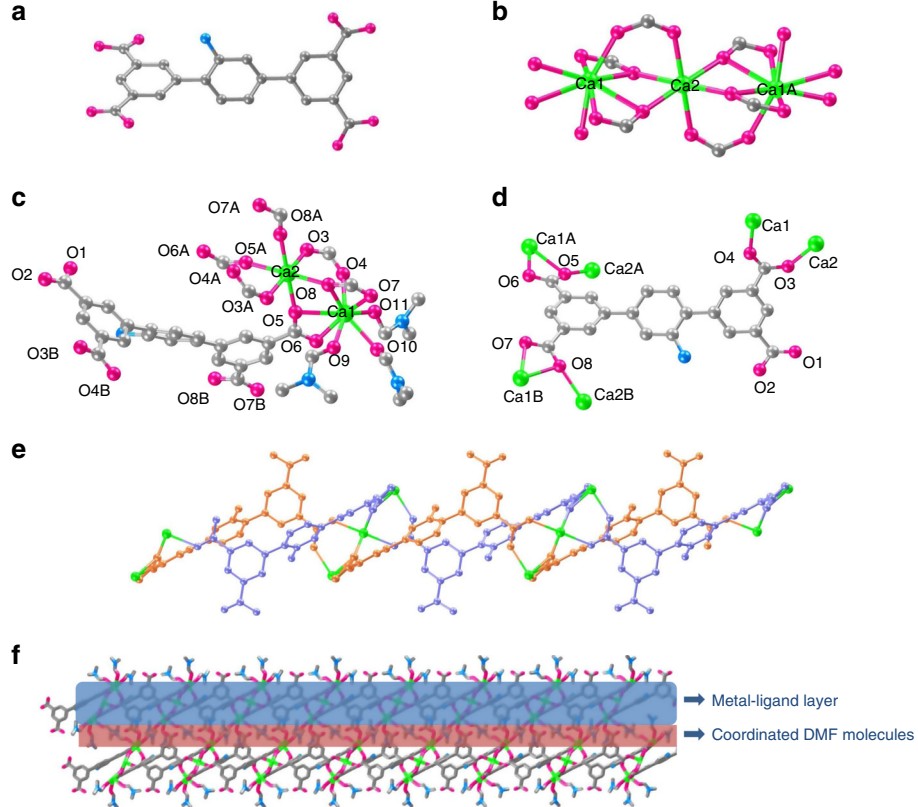

**Fig. 1** Crystal structure of Ca–MOF. $HL^{3-}$ ligand (**a**), {$Ca_3O_{18}$} metal-oxygen cluster (**b**), coordination environment of Ca1 and Ca2 ions (**c**), coordination mode of $HL^{3-}$ ligand (**d**), Z-like metal-ligand chain viewed along b axis (**e**, DMF molecules are omitted for clarity), and view of 2D metal-ligand coordination layers along b direction (**f**). Atom color: Calcium, green; Carbon, gray; Oxygen, pink; Nitrogen, blue; Hydrogen atoms are omitted for clarity. For **e**, orange and purple colors dictate atoms from different ligands

dependent shifting tendency. For high energy excitations ($\lambda_{ex} = 265$–408 nm), a strong emission peak can be observed shifting from 430 to 470 nm gradually. Further lowering the excitation energies ($\lambda_{ex} = 420$–450 nm) will result in depression of this emission peak, and a long wavelength emission shoulder above 550 nm becomes distinct, manifesting dual-emission combination character. At excitation wavelengths longer than 450 nm ($\lambda_{ex} = 460$–530 nm), the low energy emission above 550 nm becomes dominant, which also shows steady wavelength shifting (Fig. 2b).

Such kind of continuously excitation-dependent shifting emissions have been rarely reported in MOF materials. According to the dual-emission peaks we assigned for $H_4L$ ligand and the special 2D layered structure of Ca-MOF, we propose a dual-channel emitting pathways. In which, the high energy emission below 500 nm should be related with interlayer trapped excitons by the 2D layers after interligand charge transfer, and the low energy emission above 500 nm can be ascribed to the intralayer formed excimers[35]. Time-resolved PL measurements of Ca-MOF were tested (Supplementary Fig. 6), and the lifetime was determined to be 2.5 (Supplementary Table 2) and 1.0 ns for 480 and 560 nm, respectively. Difference in the two decay lifetimes supports the above dual-channel emitting mechanism. Uniquely, the distinct excitation-dependent shifting tendency of both emissions in Ca-MOF bulk crystals resembles the emitting property of carbon dots or related materials with semiconductor nature[44, 45]. This may be due to the fact that the 2D layered structure in Ca-MOF with spatial anisotropy leads to a dimensional confinement effect, forming trapped excitons and excimers with multiple energy levels, which are subject to the packing and defect states of crystal structures. Light-polarization-dependent PL spectra of Ca-MOF crystal were also measured (Supplementary Fig. 7). The direction of light polarization was

adjusted by rotating the half-wave plate, while the sample was fixed on the quartz glass. When excited at 405 nm, the observed interlayer emission intensity increased steadily by varying the polarization angle from 40 to 190°. While excited at 460 nm, the observed intralayer emission intensity increased steadily by varying the polarization angle from 40 to 160°, following by a reduction at 190°. These two different light-polarization-dependent PL provide evidence for the intra-layer and inter-layer emissions, respectively, although more thorough and direct evidence is still waiting for exploration[35]. As a result, the 2D Ca-MOF bulk crystals show an overall emission color shifting from blue to yellow and then red region with continuously changing the excitation wavelengths, as dictated by CIE coordinates (Supplementary Fig. 8).

**Ultrasound exfoliation or morphology transformation**. As discussed above, the layered structure in 2D Ca-MOF crystals sustained by van der Waals interactions between DMF molecules might be delaminated to result in exfoliation or different morphology transformation. To test this, we first study the liquid sonication of Ca-MOF in different solvents ($H_2O$, or DMF/$H_2O$ with different ratios). Tyndall effect was observed through the supernatant of Ca-MOF sonicated in pure water (Supplementary Fig. 9), indicating that the bulk crystals have been dispersed into nano-scale particles. Further SEM measurement proved the morphology of Ca-MOF is turned from bulk crystals to nanobelts with length in μm scale (Fig. 3b). AFM image shows two randomly selected nanobelts with heights of 11.1 and 21.2 nm, and widths of 140 and 120 nm, respectively (Fig. 3c, d). This proves the successful result of liquid exfoliation from bulk crystals to nano-scale morphologies. PXRD patterns display the disappearance of most diffraction peaks for the nanobelt samples

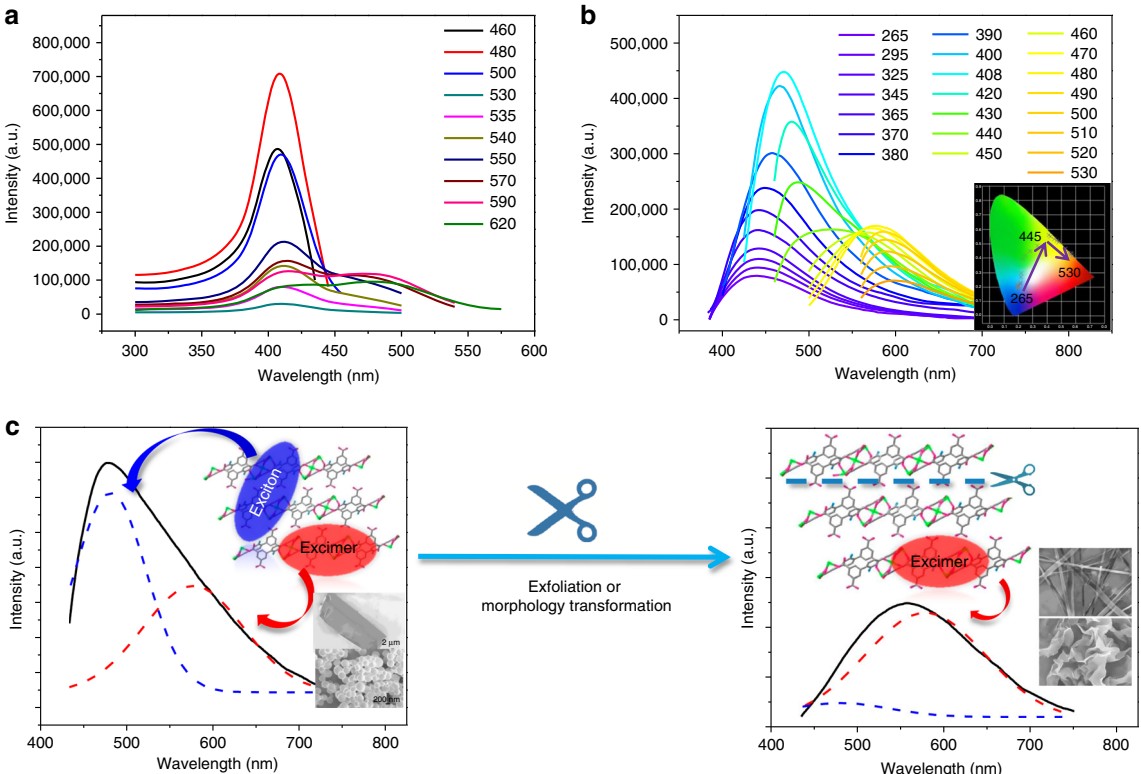

**Fig. 2** Luminescence tailoring. Excitation (**a**), and excitation-dependent emission (**b**) of 2D Ca-MOF bulk crystals (inset shows the shifting tendency of CIE coordinates with different excitation wavelengths). **c** Schematic representation of the dual emission pathways (shown in peak-deconvolution manner) of Ca-MOF bulk crystal and nanosphere samples: interlayer excitons-related emission (blue dash) and intralayer excimers-related emission (red dash), and tuning off the former by cutting down layered structures in nanosheet and nanobelt samples by exfoliation and morphology transformation

compared with that of Ca-MOF bulk crystals, showing the long-range ordered 2D layered structure has been cut down. The following result negated that the MOF structure was decomposed into inorganic salt and ligand, since there was no precipitation observed after excessive $CO_3^{2-}$ was added into the supernatant after exfoliation, suggesting that no free $Ca^{2+}$ ions were dissociated. Furthmore, when $D_2O$ was used as solvent during exfoliation process, no ligand-based proton peak was found in the supernatant by $^1H$ NMR measurement (Supplementary Fig. 10 and Supplementary Methods).

As we described in crystal structure analysis, the pendent DMF molecules in Ca-MOF are labile, which we speculate that might be replaced by water, and responsible for the morphological transformation from bulk crystals to nanobelts. TG-MS (Supplementary Fig. 11) and EA (Supplementary Table 3) analyses proved this hypothesis. In general, the weight loss contours for bulk crystals and nanobelt samples are resembling for the main framework section (corresponding to the weight loss from ~75% to ~10%), showing that the intrinsic metal-ligand coordination skeletons are maintained during the morphological transformation. However, the loss of DMF molecules detected for bulk crystals becomes inconspicuous for nanobelt samples obtained by ultrasound in water, while an additional weight loss for $H_2O$ molecules can be detected before 150 °C, manifesting the replacement of DMF by $H_2O$ molecules. This replacement results in breakdown of the long-range packing 2D layer structures sustained by weak van der Waals interactions among DMF molecules in Ca-MOF bulk crystals, and the macroscopic morphologies are simultaneously transformed into nanobelts. In order to make clear the coordination structure for the obtained nanobelt samples, Atomic Pair Distribution Function (PDF) tests were conducted to be compared with that of the bulk crystal, which is a powerful technique to obtain the local coordinate

structure of materials, and can show fingerprint character for the specific environment in the interested coordinate region[46]. As shown in Supplementary Fig. 12, for the local coordination environment around Ca(II), the peaks at about 2.4 Å are similar for both the nanobelt and bulk crystal samples, relating to the Ca–O coordination bonds of Ca(II) with the ligands and DMF or water molecules, manifesting that the basic interplane metal-organic coordination structures are maintained in the nanobelt. Furthermore, at around 4 Å, another peak appears for the nabobelt, which is absent for the bulk crystal. According to the single crystal data obtained for Ca-MOF-$H_2O$ as discussed below, there exist uncoordinated $H_2O$ molecules in the crystal lattice of the water-changed samples, and the Ca–O distance from Ca(II) center to the nearest lattice $H_2O$ molecule is around 4 Å, supporting the appearance of the 4 Å peak in the PDF curve we obtained here. This further proves the substitution of DMF by water in the nanobelt sample, and in consistent with the TG and EA results as well. As an alternative proof that DMF molecules in the Ca-MOF could be replaced by water while maintaining the metal-ligand layer, single crystals of Ca-MOF was soaked in methanol solvent containing little amount of water (2%, volume ratio) for 24 h, water-substituted crystals of Ca-MOF-$H_2O$ were obtained as determined by SCXRD (Supplementary Table 1). The metal-oxygen center and metal-ligand layer structure is quite similar to that of Ca-MOF, while Ca1 of Ca-MOF–$H_2O$ shows a seven-coordinated mode attached by two water molecules, compared to the eight-coordinated mode in Ca-MOF with DMF coordination (Supplementary Fig. 13). This is in support of our assumption, that exfoliation and structural transformation of the Ca-MOF into different morphologies can be induced during the ultrasonication. A two step process might be

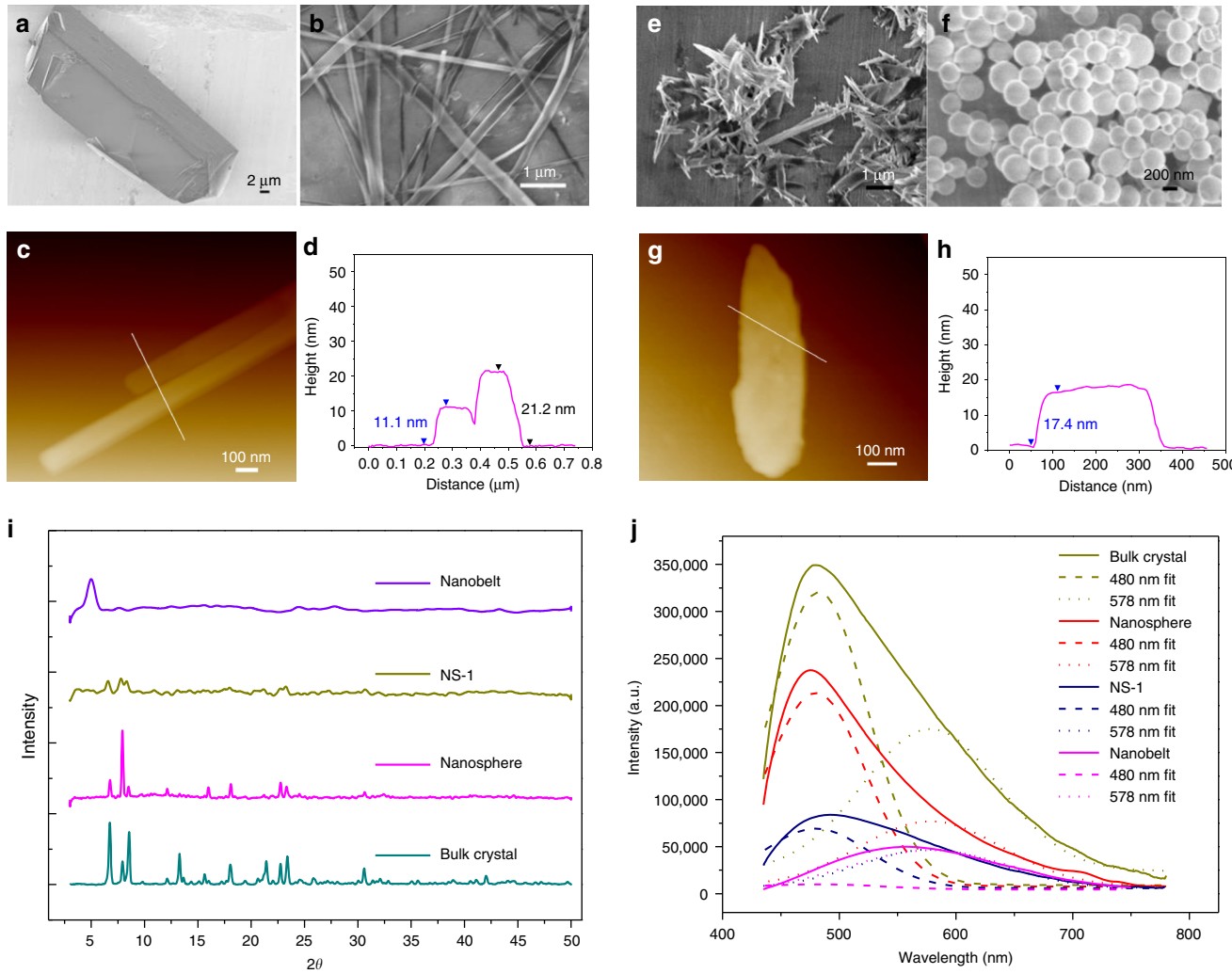

**Fig. 3** Ultrasound exfoliation. SEM images of as-synthesized 2D Ca-MOF bulk crystals (**a**), and the obtained nanobelts after ultrasound exfoliation in H$_2$O (**b**). AFM images (**c**) and the corresponding height profile (**d**) of two randomly selected nanobelts. SEM images of the obtained nanosheets (NS-1) (**e**) and nanospheres (**f**). AFM image (**g**) and height profile (**h**) of NS-1. PXRD patterns (**i**) and emission spectra (**j**, shown in peak-deconvolution manner by dash lines, $\lambda_{ex} = 408$ nm) of Ca-MOF bulk crystals, nanobelts, nanospheres and NS-1 samples

involved, replacement of DMF by water, and then breakdown of the layer structures.

Furthermore, we found that the obtained nanobelts will transform into another phase with inhomogeneous grass-like nanosheet morphology (named as NS-1) by sonication in DMF/H$_2$O (v/v = 5/2). NS-1 restores some of the PXRD peaks of Ca-MOF bulk crystals (Fig. 3i), manifesting the partial recovery of the original 2D ordered structure. However, SEM image displays sheet-shaped morphology for NS-1 (Fig. 3e), different from the bulk crystals and nanobelts. AFM image shows that the randomly selected NS-1 sample exhibiting a thickness of 17.4 nm (Fig. 3g, h), which is composed of about sixteen metal-ligand coordination layers, as each layer is estimated to be ~10.98 Å based on the crystal structural analysis of Ca-MOF.

As shown in the above ultrasonic transformation from bulk crystal to nanobelt and nanosheet (NS-1) samples, we can see that the solution of H$_2$O and DMF plays an important role. To further prove this, mixed DMF/H$_2$O solution with different ratios were used for the sonication process. SEM images (Supplementary Fig. 14) show quite different morphologies depending on water ratio of DMF/H$_2$O mixture. When water content was below 33% (Supplementary Fig. 14a–c), no obvious change in the morphology was observed for bulk Ca-MOF crystals but with a trend of

fracture and decreasing in the particle size. When water content reached 50%, some sphere-like samples were observed (Supplementary Fig. 14d), and when water content reached 67%, almost all of the bulk crystals were transformed to nanospheres with several hundred nanometers in diameter (Fig. 3f and Supplementary Figure 14e). When water content was up to 75%, nanobelt morphologies appeared (Supplementary Fig. 14g), accompanied by the breakdown of layered 2D structures.

Continuing experiments prove that, the above transformation among different morphologies of Ca-MOF can be happened and reversed by sonication in different DMF/H$_2$O solvents, as summarized in Fig. 4. First, bulk crystals would transform to nanobelts in pure water, and transform to nanospheres in DMF/H$_2$O solvents with 67–75% water content. Nanobelts, nanospheres, and nanosheets could transform among each other. Simply speaking, when the samples were ultrasound in DMF/H$_2$O solvents with 67–75% water content, nanospheres could be obtained. Similarly, nanobelts and nanosheets (NS-1) could be achieved by ultrasound in pure water and DMF/H$_2$O (v/v = 5/2) solvents, respectively (Supplementary Fig. 15). The readily accessible and reversible nature among different morphological transformations in DMF/H$_2$O with varied water content also prove that, no significant dissociation of the metal-organic

coordination structures should be involved during the processes, and the transformations should be directly related with the dynamically replacement between DMF and water molecules, as supported by TG-MS and EA analyses. In different morphological samples, the basic metal-organic coordination sheets will be preserved, while the long-range ordered multi-layered 2D structure might be breakdown in nanosheet or nanobelt samples, due to the reduction in the thickness dimension.

Knowing from above, nanosheet was obtained by ultrasonicating the powder of Ca-MOF. We were curious to know whether it is possible to prepare 2D layers of $[Ca_3O_{18}]$ directly by ultrasonicating a concentrate solution of the precursors with an appropriate ratio of DMF/$H_2O$. A series of ratios have been tried, when the ratio were 5/2 and 4.5/2.5, some precipitates were obtained. PXRD data indicated different crystal diffraction peaks comparing to Ca-MOF, and SEM images showed block-like morphology with large size in micron scale (Supplementary Fig. 16). Therefore, it is further proved that the exfoliation result we obtained herein was based on the formation of 2D layer structure of Ca-MOF first, and then breakdown of the layered structure due to replacement of DMF by water molecules.

**Grinding exfoliation**. As we can see, the Ca-MOF morphologies are highly water sensitive under liquid sonication conditions. Therefore, we tried other methods to see whether the bulk Ca-MOF crystals can be exfoliated and transformed, for which the samples were ground under relative humidity (RH) of 55 and 75%, respectively. At the low RH of 55%, SEM images exhibit granular morphologies even after 40 min of grinding (Fig. 5a and Supplementary Fig. 17). At high RH of 75%, plate-like nanosheets (named as NS-2) appeared after grinding for 5 min, and achieved unified morphology after 8 min (Supplementary Fig. 11e, f). XRD

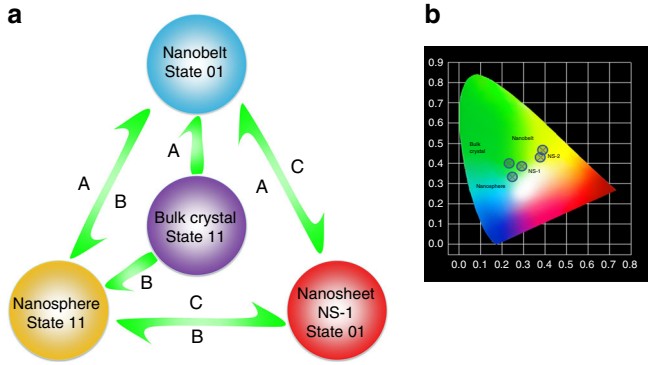

**Fig. 4** Morphological transformation. **a** Transformation relationship and optical memory states for nanobelts, nanospheres, and nanosheets (NS-1). Transformation conditions under ultrasound: A, in $H_2O$; B, in DMF/$H_2O$ (v/v = 1/2, water content: 67%); C, in DMF/$H_2O$ (v/v = 5/2). State 01 refers to interlayer emission off + intralayer emission on, and State 11 refers to interlayer emission on + intralayer emission on. **b** Emitting color tone shown by CIE coordinates

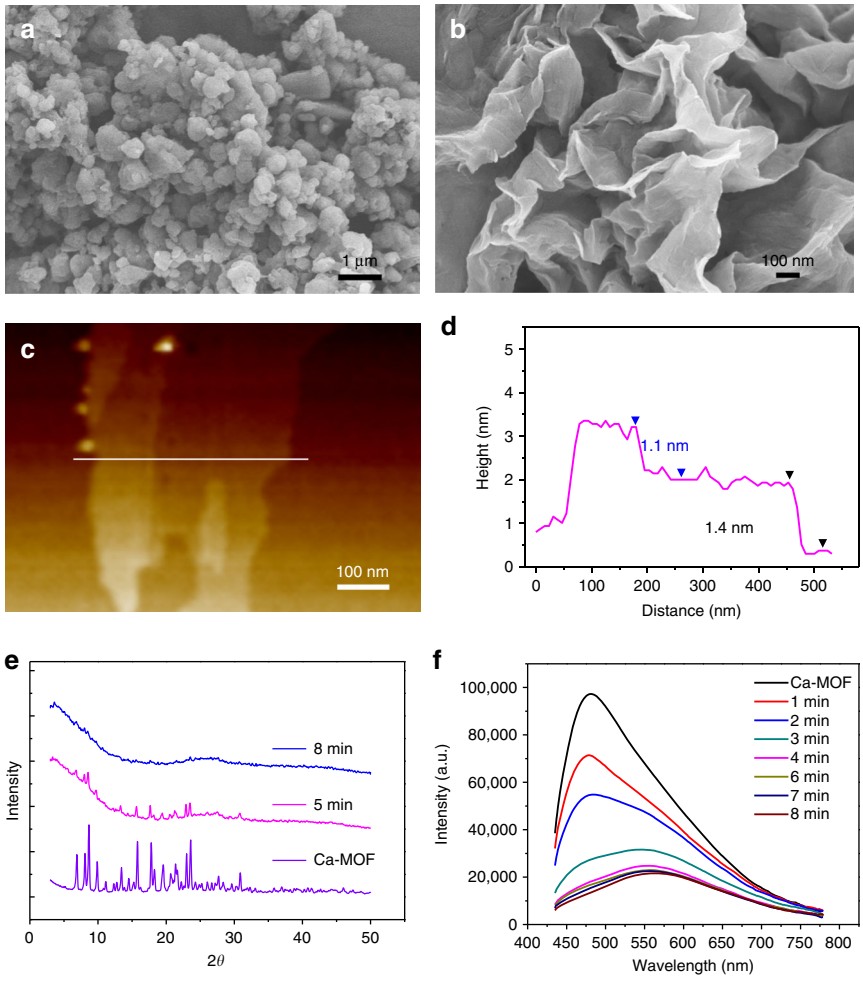

**Fig. 5** Grinding exfoliation. SEM images of Ca-MOF bulk crystal samples after grinding for 8 min at RH = 55% (**a**) and RH = 75% (**b**). AFM image (**c**) and height distribution profile (**d**) of NS-2. PXRD (**e**) and emission spectra (**f**) of Ca-MOF after grinding for different time at RH = 75%

patterns (Fig. 5e and Supplementary Fig. 18) show that the sharp diffraction peaks will diminish after transforming into the nanosheet (NS-2) states, due to the disappearance of the long-range ordered layer structure. But the basic metal-ligand coordination structure might still be preserved as proved by EA and TG-MS tests. Since the nanosheets are very thin and flexible, they show a stacked appearance (Fig. 5b). AFM images manifest two steps of 1.1 and 1.4 nm for NS-2 sample (Fig. 5c, d), near to the thickness of single metal-organic coordination layer in Ca–MOF crystal structure (~10.98 Å). To make the conclusion more credible, several NS-2 samples were taken for AFM measurements (Supplementary Fig. 19). In general, the obtained NS-2 nanosheets might have single, two or three metal-organic coordination layers in the thickness dimension, indicating the successful exfoliation of Ca–MOF bulk crystals into ultrathin nanosheets through simple and feasible grinding method. Different from the exfoliation process achieved by sonication, in which the transformation is happened on a liquid-solid interface, herein, the exfoliation is actually involved on a gas-solid interface, and greatly dependent on the water vapor content in air (humidity).

## Discussion

The emission spectra of different morphologies of Ca-MOF obtained by ultrasound and grinding methods were studied (Figs. 3j and 5f). By detailed peak deconvolution, we can see that the nanospheres and bulk crystals have almost the same peak distribution between high energy interlayer excitons-related emission and low energy intralayer excimers-related emission. This might be due to the fact that although the morphology has been changed in nanosphere samples, the ratios among scales of the three dimensions are almost unchanged compared with bulk crystals. Therefore, the two channeled emissions are not affected greatly, only with reduction in intensities due to the enhanced dissipation process after morphological transformation. However, in nanosheet samples (NS-1) with the thickness dimension greatly reduced to about 10–20 nm, peak deconvolution manifests that the intensity of interlayer emission is depressed more dramatically, while that of the intralayer emission is reduced only to a small extent. Therefore, the overall emission color of the NS-1 samples is shifted to near white light compared with the green emission of bulk crystals (Supplementary Fig. 20). Since the basic coordination structure in NS-1 is similar with the bulk crystals as proved by PXRD, TG-MS and EA tests, this emission change should be related with the morphological transformation in which the layer numbers are greatly cut down in the thickness dimension, and thereof hamper the interlayer excitonic emission. In nanobelt samples, the high energy interlayer excitonic emission is almost totally diminished, and the overall emission is red-shifted to yellow region (maximized at ~560 nm), with dominant contribution from the excimers-related intralayer emission. This should be due to the fact that the DMF-sustained multi-layered structure is no longer existed in nanobelt samples with low thickness, and the interlayer emitting channel is therefore tuned off.

The emission spectra of exfoliated samples obtained during solid state grinding for different time clearly manifest the steady decrease of high energy excitons-related emission at 480 nm, and dominance of low energy excimers-related emission at 560 nm in the final exfoliated samples (Fig. 5f). This further validates the speculation that the interlayer excitons-related emission can be gradually suppressed due to continuous delaminating of the multi-layered structure, reaching totally tune-off in the few layered or even single layered 2D MOF nanosheets (NS-2). As a result, the overall emitting color of the samples obtained by

grinding for different time steadily changes from green to yellow region in CIE coordinates (Supplementary Fig. 21).

The above successful manipulation of dual-channel emission in Ca-MOF by reversible exfoliation and morphology transformation prompts the conceptual application as a writing-reading-erasing type optical memory, for which the writing and erasing process can be realized by tuning on/off the interlayer excitonic emission. As shown in Fig. 4, the memory states of 01 and 11 can be swapped alternately by facile sonication in different solvents to obtain certain morphological samples. Similarly, grinding process can also be applied to achieve switchable optical states under varied humidity conditions. The balance between robustness and dynamics of the 2D-MOF structure herein warrants reproducibility for the fabrication and operation of the proposed supramolecular-based optoelectronic devices. The working theory of the devices might be relied on the switching of different states of the samples using such methods of sonicating, grinding or rubbing under humid conditions. Although in the present stage, it is still of impracticalities to make a real working device based on existing experimental conditions and techniques. But with further and deeper exploration along this line, we hope to achieve the final goal of device fabrication and operation in the near future.

In summary, a unique 2D Ca-MOF was obtained, which possesses a multi-layered coordination structure with balanced robustness and dynamics. Exfoliation and morphological transformation can be achieved by either ultrasound via solid-liquid interface, or grinding via solid-gas interface, leading to ultrathin nanosheet or nanobelt samples with greatly reduced magnitude in the thickness dimension. Interestingly, the 2D layered structure of Ca-MOF affords interlayer excitons-related and intralayer excimers-related dual-channel emissions. Due to the reversible formation/breakage of multi-layered structure during morphological transformations, the interlayer excitation pathway can be tuned on/off to manipulate the overall emitting color and optical memory states for different Ca-MOF samples. This develops a new concept to synergetically combine bottom-up metal-organic coordination structural design and top-down morphological modification in 2D-MOFs, to achieve new photo-electronic prospects and pave the way for applications in advanced photonic fields.

## Methods

**General**. All reagents and solvents were commercially available and used without further purification. FT-IR spectra were recorded on a Nicolet/Nexus-670 spectrometer in the spectral range 4000-400 cm$^{-1}$ using the KBr disk. The $^{1}$H-NMR spectra were recorded on a Bruker Avance 400 NMR spectrometer, applying TMS as the internal standard and using MestReNova software to analyze the spectra. The UV-vis absorption spectra were measured on a SHIMADZU UV-3600. Photoluminescence spectra were measured on EDINBURGH FLS980 fluorescence spectrophotometer. TG and TG-MS spectra were carried out on a NETZSCH TG209 instrument under a nitrogen flow with a heating rate of 10 °C/min. PXRD data were recorded on a Rigaku SmartLab X-ray diffractometer with Cu–Kα radiation ($\lambda$ = 1.54056 Å) at room temperature. SCXRD data were collected on a Rigaku Oxford SuperNova X-RAY diffractometer system equipped with a Cu sealed tube ($\lambda$ = 1.54178 Å) at 50 kV and 0.80 mA at 150 K (Supplementary Data 1 and 2). AFM measurements were carried out on Dimension FastScan atomic force microscope. SEM measurements were performed on Hitachi SU8010 scanning electron microscope. Al foil was used as substrate in SEM measurement, and Si plate was used in AFM measurement. Polarization-dependent PL measurements were measured by Horiba iHR 320 equipped with Continuum SuliteII laser excitation applying a rotatable half-wave plate. PDF (Atomic Pair Distribution Function) data was collected using mython linear detector (1 K) at SSRF (Shanghai Synchrotron Radiation Facility) 14B beamline. The powder samples were sealed in the glass capillary (0.01 mm wall thickness, 1.0 mm outer dameter). The mython detector is fixed on the diffractometer arm about 0.75 meter from the diffractometer center. Large two-theta scanning (2–80°) mode was used with X-ray energy of 18 keV. The measurement time for each two-theta angle is 100 s and the two-theta step is fixed at around 3.906 degree. Q resolution for the whole Q range is better than 0.01 in this experimental setup. The data were processed by using software PDFgetX3S2[47].

**Synthesis of [Ca₃(HL)₂(DMF)₅]ₙ (Ca-MOF).** H₄L was prepared in a similar method according to the reference[48] and characterized by $^1H$ NMR (400 MHz, DMSO-$d_6$, 25 °C, δ): 13.34 (s, 4 H), 8.45 (d, 2 H), 8.41 (s, 2 H), 8.24 (s, 2 H), 7.24 (s, 1 H), 7.20 (d, 2 H), 7.06 (d, 2 H), 5.25 (s, 2 H). Twenty milligram (0.047 mmol) H₄L and 20 mg (0.18 mmol) $CaCl_2$ was added into a 20 mL glass bottle, then 6 mL DMF/H₂O (v/v = 5/2) mixed solution was injected following by about 2 min ultrasound for clear yellow-green mixture. Sealed by a cover, the glass bottle was heated at 85 °C for 48 h. The reaction mixture was then allowed to cool to room temperature naturally. Colorless block-shaped crystals of Ca-MOF were collected, washed with DMF/H₂O (v/v = 5/2) solution for three times and dried under vacuum for 10 h. Yield 55%, based on H₄L. Anal. calcd for $C_{59}H_{59}N_7O_{21}Ca_3$: C, 53.59; H, 4. 46; N, 7.41. Found: C, 53.12; H, 4.54; N, 7.28. FT-IR (KBr, $cm^{-1}$): $\nu$ = 3340 (w), 2927 (w), 1708 (m), 1656 (s), 1608 (m), 1556 (m), 1382 (s), 1214 (m), 1108 (m), 775(m), 732(m), 678 (w). TG curve of Ca-MOF shows a weight loss of 1.2% before 100 °C, which should be attributed to a small amount of water from surrounding environment. There is no obvious weight loss before 230 °C, and the two obvious weight loss between 230 and 530 °C should be ascribed to the loss of DMF molecules with a weight loss of 27.4%. The overwhelming weight loss after 530 °C indicates the decomposition and collapse of the framework.

**Transformation of Ca-MOF into Ca-MOF-H₂O {[Ca₃(HL)₂(H₂O)₄]·4H₂O}ₙ.** Five milligram Ca-MOF crystals and 6 mL methanol/water solution (water content 2%, volume ratio) were mixed in a 20 mL glass bottle. The bottle was sealed and heated to 85 °C for 24 h. Then the crystals were separated and the structure was determined by SCXRD under sealing conditions.

**Exfoliation and morphology transformation by ultrasound.** In a typical experiment, 5 mg Ca-MOF was dispersed in 10 mL different solvents (H₂O, or DMF/H₂O with different ratios, respectively) and sonicated in a KQ118 ultrasonic instrument (70 W, Kun Shan Ultrasonic Instruments Co.) for 20 min (as the ultrasound would produce considerable heat during the experiment, cleaning fluid water should be replaced every 5 min). Then the sample was allowed to stand for precipitation, and the supernatant was used for scan electron microscopy (SEM) or/and atomic force microscopy (AFM) study. Nanobelt sample was obtained from ultrasonic treatment in H₂O and collected by centrifuging, and nanosphere was obtained from treatment in DMF/H₂O (v/v = 1/2, water content: 67%) mixed solution. Nanobelt sample could be transformed to nanosheet (NS-1) in DMF/H₂O (v/v = 5/2) mixed solution under ultrasound.

**Exfoliation by grinding.** Five milligram Ca-MOF bulk crystals were added into a mortar with 10-mm in diameter and ground with a pestle under different humidity. The value of relative humidity was monitored by a hygrometer. The obtained samples were separated and dispersed in DMF solvent. Then the diluted dispersion was stand still, and the supernatant was dipped on flat substrate for SEM measurement.

**Data availability.** The data that support the findings of this study are available from the corresponding author on request. The X-ray crystallographic coordinates for structures reported in this study have been deposited at the Cambridge Crystallographic Data Centre (CCDC), under deposition numbers 1584130 and 1823260. These data can be obtained free of charge from The Cambridge Crystallographic Data Centre via www.ccdc.cam.ac.uk/data_request/cif.

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

## Acknowledgements

The authors thank the help from SSRF 14B beamline scientists and Prof. Huaidong Jiang from ShanghaiTech University for the PDF tests. We acknowledge the financial support of NSFC (21771197, 21720102007, U1732120), NSF of Guangdong Province (S2013030013474), FRF for the Central Universities, and STPP of Guangzhou (201504010031).

## Author contributions

M.P. designed the research and wrote the paper. W.-M.L. carried out most of the syntheses and measurements. J.-H.Z., S.-Y.Y., H.-P.W., K.W., Z.W., Y.-N.F., and C.-Y.S. helped in some experiments and data analyses. H.L., X.Z., and J.W. conducted the PDF tests and analyses. All authors discussed the results and commented on the manuscript.

## Additional information

**Competing interests:** The authors declare no competing interests.

