## [Peer Review File · Nature Communications]

Reviewers' comments:

Reviewer #1 (Remarks to the Author):

Liao et al. report the synthesis and characterization of a new layered Ca-based Metal Organic Framework (MOF). The two dimensional structure of the MOF is composed of infinite layers of [Ca3O18] which are capped by solvent molecules (DMF and/or H2O). Upon exfoliation of the [Ca3O18] layers and substitution of DMF with water molecules, the bulk phase can convert to various morphologies, such as nanospheres, nanobelts, and nanosheets. The conversion between the different morphologies is reversible but not the transformation from bulk to the various nano-morphologies. The isolated crystal morphologies exhibit interesting emission properties which can potentially be used to identify district optical memory states.

Manuscript is well written and authors have performed some basic characterization. The structural characterization of the various nano-morphologies needs improvements. I would recommend publication to Nature Communications after some major revision. In more details:

The authors speculate the same metal-ligand coordination environment between the bulk compound and the different nano-morphologies based on the optical absorption (EA) and TGA-MS data. Unfortunately, neither of these two techniques is the right probe for characterizing the local structural at the atomic scale. I believe characterization of the structure of the nano-compounds is crucial for understanding and controlling the interesting optical properties of these materials. Structural characterization with local probes such as Extended X-ray absorption fine structure (EXAFS) or even better Pair Distribution Function (PDF) analysis must be performed. Only then, some conclusions about the similarity or difference between the structure of bulk and nano can be drawn.

The extended structure of the pristine Ca-MOF does not seem to play an important role to the properties. Have the authors tried to ultrasonicate a concentrate solution of the precursors with an appropriate ratio of DMF/H2O as an attempt to form directly the 2D layers of [Ca3O18]? This procedure could cut down on the overall synthesis time.

Since the amount and type of solvent molecules around the 2D layers have a large effect on the morphology of the particles, it will be interesting to see what the pristine morphology is when no solvent is present. A typical activation step in MOF chemistry is the utilization of Critical Point Drying for solvent removal. Is it possible to apply this preparation method in this material?

Reviewer #2 (Remarks to the Author):

Interesting results! But there are many issues as stated below:

- 1) The exfoliation with water might cause the MOF to decompose to ligand. Although they mention that "The following result negated that the MOF structure was decomposed into inorganic salt and ligand, since there was no precipitation observed after excessive CO₃²⁻ was added into the supernatant after exfoliation, suggesting that no free Ca²⁺ ions were dissociated. " They did not test for organic ligand via NMR, in which, they can sonicate the MOF in D₂O and run NMR.
- 2) Why the authors used the term "excimer". Accordingly to Ref. [35], it should also be an exciton formed by excited electron cloud within the layer. The term "excimer" has a different meaning.
- 3) To identify the emission peaks as a result of interlayer and intralayer exciton recombination, the authors should carry out time-resolved PL measurements, as described by Ref. [35], which could eliminate the possibility of emission by ligands.
- 4) Furthermore, the authors should conduct light-polarization-dependent PL measurements to provide direct evidence for intra-layer and inter-layer exciton emission, see the details in Ref. [35].

5) Referring to the discussion section, the authors mentioned that " ... dual-channel emission ... prompts the conceptual applications of ... optical memory." The concept of "optical memory" is referred as to a material which exhibits a different optical property after being exposed to laser excitation. Here, the authors did not explain how their MOFs can be switched to have different emission properties after being exposed to a laser beam.

Reviewer #3 (Remarks to the Author):

In this paper the authors report a new Ca-MOF made using a fluorescent ligand, and which is because of this ligand fluorescent itself. This fluorescent MOF emits at two different wavelengths and the intensity of one of these emissions can be changed by sonicating the MOF in different solvents producing new nanoscale morphologies. The ability to tune the optical properties of a material through the formation of nanoscale phases is very topical, with many examples in purely inorganic materials (e.g. graphene, MoS₂), but this phenomenon has been rarely explored in MOFs. CaMOFs are relatively unusual, and the apparent stability of this material is itself quite exciting. The analysis of the data is good throughout, at the level which it has been carried out. The results in this paper are very intriguing but sometimes there could be more thorough interpretation and explanation with less speculation on properties. The two questions which seem unclear to me are:

What is the nano belt phase? It is described as a delaminated version of the Ca-MOF, but the powder pattern unambiguously proves it is not closely related to the Ca-MOF in terms of structure. There may not be free Ca ions, but this says nothing about the structure of the material. All explanations of the fluorescence of this material on the basis of particular structural changes are therefore unconfirmed. The authors do hint at the different structure of this material, but I think an upfront description of the change is necessary: to produce nanosheets of the Ca-MOF material through ultrasonication is a two step process and the authors should be clear about this in text and figures. It would be interesting if the authors could suggest a structure for this phase, perhaps backed up through either surface sensitive diffraction measurements or high resolution microscopy.

(2) The description of the two kinds of emission could be clarified and the origin of the morphology-dependence is a little unclear. Solution measurements show that on reduced aggregation the lower energy band disappears, but in the solid state it is this lower energy band that remains on reduction of dimensionality. This is not the usual behaviour and the contrast between the solution and solid state data is not fully explained. The assignment of the 'interlayer' vs 'intralayer' bands seems to be somewhat circular (the band that disappears on delamination must be interlayer; the interlayer band disappears because of delamination) which means the interpretation of the physical mechanism for the observed changes is not justified. Why should a band that appears for ligands dissolved at low concentrations be assigned to interlayer emission in the solid? If this could be more clearly explained it would be very useful. A more complete explanation of these critical data would greatly benefit this paper

Other issues:

The spectroscopic fitting of the datasets should be able to yield quantitative information, but I can't seem to find this information reported. It would be very useful to understand the datasets: particularly as some figures (e.g. Fig 2b) are extremely difficult to interpret. For Fig 2b in particular, perhaps a 2D data plot would be better? Or at least a consistent coloring of the line corresponding to the incident radiation (e.g. blue for low wavelength to red for high wavelength).

The proposed application is not very novel and seems somewhat unsuitable: the transformations described by the authors require ultrasonication and large scale morphological changes. This would be an extremely unwieldy memory device. This material could be useful for sensing or similar applications, but the proposed mechanism seems to be a long way from a realisable memory

device.

More minor issues:

Sometimes the choice of words is not quite appropriate
'scissoring' is not really a physical description for the delamination of these materials
'peak splitting' should be replaced by peak deconvolution or similar

Please report crystal size and absorption correction method in the CIF, and refinement strategy briefly in the SI.

In conclusion, this is a very interesting material, which undergoes a number of unusual chemical transformations with interesting optical properties. However, as it stands, the paper does not fully explain the chemical behaviour or the optical properties, which reduces its value significantly. Were this to be done, I would recommend this paper for publication.

Response to Reviewers' comments:

Based on the reviewers' comments, we carefully revised our paper and the point-to-point responses to the reviewers' comments were listed in the separated pages. All changes in the manuscript are highlighted in **RED**.

To Reviewer #1

Comments:

Liao et al. report the synthesis and characterization of a new layered Ca-based Metal Organic Framework (MOF). The two dimensional structure of the MOF is composed of infinite layers of $[\text{Ca}_3\text{O}_{18}]$ which are capped by solvent molecules (DMF and/or H_2O). Upon exfoliation of the $[\text{Ca}_3\text{O}_{18}]$ layers and substitution of DMF with water molecules, the bulk phase can convert to various morphologies, such as nanospheres, nanobelts, and nanosheets. The conversion between the different morphologies is reversible but not the transformation from bulk to the various nano-morphologies. The isolated crystal morphologies exhibit interesting emission properties which can potentially be used to identify district optical memory states. Manuscript is well written and authors have performed some basic characterization. The structural characterization of the various nano-morphologies needs improvements. I would recommend publication to Nature Communications after some major revision. In more details:

1. The authors speculate the same metal-ligand coordination environment between the bulk compound and the different nano-morphologies based on the optical absorption (EA) and TGA-MS data. Unfortunately, neither of these two techniques is the right probe for characterizing the local structural at the atomic scale. I believe characterization of the structure of the nano-compounds is crucial for understanding and controlling the interesting optical properties of these materials. Structural characterization with local probes such as Extended X-ray absorption fine structure (EXAFS) or even better Pair Distribution Function (PDF) analysis must be performed. Only then, some conclusions about the similarity or difference between the structure of bulk and nano can be drawn.

Response: Many thanks for the reviewer's valuable suggestion. Indeed, EXAFS is a good technique to probe detailed structure of the nano-materials. Once we received this comment, we started to apply for an EXAFS test from Shanghai Synchrotron Radiation Facility and also consulted to other possible resources from our collaborators in France and Australia. But unfortunately, due to the scarce availability and long queue, we can not have a chance to do EXAFS test in the near future.

Alternatively, we tried another method, by soaking the single crystals of the Ca-MOF (LIFM-41) in methanol solvent containing little amount of water (2%, volume ratio) for 24 hours, water-substituted crystals Ca-MOF- H_2O were obtained. In the structure of Ca-MOF- H_2O (Figure S12), we can see that the originally coordinated DMF molecules are substituted by H_2O molecules, while **the metal-ligand layer is** basically intact. This is in support of our assumption, that the exfoliation and structural transformation of the Ca-MOF into different morphologies can be induced by replacement of the coordinated DMF molecules, as detected by the EA and TG data summarized in Table S3. It is also noted that the transformation of

Ca-MOF between different morphologies is facily reversible (Figure 4). This also proves that the coordination environment around the Ca(II) centers during morphological transformation processes should not be too dramatic, and the assumption of DMF substitution is plausible. The corresponding experimental result and structure of Ca-MOF-H₂O have been added in the text and SI.

2. The extended structure of the pristine Ca-MOF does not seem to play an important role to the properties. Have the authors tried to ultrasonicate a concentrate solution of the precursors with an appropriate ratio of DMF/H₂O as an attempt to form directly the 2D layers of [Ca₃O₁₈]? This procedure could cut down on the overall synthesis time.

Response: Thanks for the reviewer's kind suggestion. We have tried to synthesize the 2D layers of [Ca₃O₁₈] directly by ultrasonating the concentrate solution of the precursors with various ratios of DMF/H₂O, but it failed. In detail, 0.22 g (2 mmol) CaCl₂ powder was dissolved in water (1, 1.4, 2, 2.5, or 3 mL), and added into DMF solution (6, 5.6, 5, 4.5, or 4 mL) including 0.21 g (0.5 mmol) H₄L ligand. Then the mixture was sonicated in ultrasonic instrument for 30 min. When the ratio of DMF/H₂O was 5/2 or 4.5/2.5, some off-white precipitates appeared (Fig. A), and the precipitates were separated for PXRD measurement (Fig. B). The diffraction peaks of these two precipitates are quite similar to each other but different from Ca-MOF, suggesting different crystal structures. Further, SEM images of these two precipitates showed block-like morphology with large size in micron scale (Fig. C). So we failed to synthesize the 2D [Ca₃O₁₈] layers of Ca-MOF directly by ultrasonating the concentrate solution of the precursors with various ratios of DMF/H₂O. This comparison experiment has been added in the text and SI.

Fig. A The mixtures obtained after ultrasonating the concentrate solution of the precursors with various ratios of DMF/H₂O.

Fig. B The PXRD patterns of Ca-MOF and precipitates obtained from ultrasonicing the concentrate solution of the precursors with different ratios of DMF/H₂O (4.5/2.5 or 5/2).

Fig. C SEM images of precipitates obtained from ultrasonicing the concentrate solution of the precursors with different ratios of DMF/H₂O, (a) for 5/2 and (b) for 4.5/2.5, respectively.

3. Since the amount and type of solvent molecules around the 2D layers have a large effect on the morphology of the particles, it will be interesting to see what the pristine morphology is when no solvent is present. A typical activation step in MOF chemistry is the utilization of Critical Point Drying for solvent removal. Is it possible to apply this preparation method in this material?

Response: Thanks for the reviewer's opinion. CO₂ Critical Point Drying was applied to bulk crystals, NS-1, nanobelts and nanospheres for 48 h. And the morphologies were obtained from SEM measurement. The result indicated that the morphologies showed no obvious change after CO₂ Critical Point Drying, though the particle size was reduced.

Fig. D SEM images of bulk crystals, NS-1, nanobelts and nanospheres after 48-h CO₂ Critical Point Drying.

To Reviewer #2

Comments:

Interesting results! But there are many issues as stated below:

1) The exfoliation with water might cause the MOF to decompose to ligand. Although they mention that "The following result negated that the MOF structure was decomposed into inorganic salt and ligand, since there was no precipitation observed after excessive CO_3^{2-} was added into the supernatant after exfoliation, suggesting that no free Ca^{2+} ions were dissociated." They did not test for organic ligand via NMR, in which, they can sonicate the MOF in D_2O and run NMR.

Response: Thanks for the reviewer's advice. In order to prove the ligand didn't decompose during the exfoliation process, D_2O was used for ultrasound solvent. In detail, 10 mg H_4L ligand was added into 2 mL D_2O and the mixture was sonicated for 20 min. As the ligand could not dissolve in water, 2 mL d_6 -DMSO was used to wash the solid after ultrasound. The supernatant was separated by centrifuge and filtered with 0.2 μm filter membranes for three times (to exclude the effect of nanoparticles). The obtained filtrate was conducted for ^1H NMR and compared with H_4L ligand. It showed that only the peak of DMF molecule (7.95, 2.89 and 2.73 ppm) was found in the filtrate (Fig. E). No other ligand-based peak was observed between 6.8 and 8.5 ppm. This result could prove that Ca-MOF did not decompose to produce ligand during the exfoliation process in water. This proof has been added into SI.

Fig. E ^1H NMR spectra of H_4L ligand (a) and the filtrate after Ca-MOF exfoliation (b).

2) Why the authors used the term "excimer". Accordingly to Ref. [35], it should also be an exciton formed by excited electron cloud within the layer. The term "excimer" has a different meaning.

Response: Before coordination, the existence of direct coupling of the ligand molecules' excited and ground states results in excimer emission above 500 nm, as supported by the fact that this emission peak appeared in high concentration solution but disappeared in low concentration (Figure S4). This emission is preserved in Ca-MOF after coordination, since coupling of the ligand molecules' excited and ground states can still happen within the layers of the solid state layered structures. And therefore, we assign the low energy emission above 500 nm in Ca-MOF to be resulted from the ligands' excimer emission within the layer.

3) To identify the emission peaks as a result of interlayer and intralayer exciton recombination, the authors should carry out time-resolved PL measurements, as described by Ref. [35], which could eliminate the possibility of emission by ligands.

Response: As the reviewer suggested, time-resolved PL measurements were tested (Fig. F), and the lifetime was determined to be 2.5 and 1.0 ns for 480 and 560 nm, respectively. The blue PL at 480 nm should be ascribed to interlayer trapped exciton on crystal or surface defects. And the PL at 560 nm independent on crystals defects should result from the intralayer excimer emission of ligand.

Fig. F Time decay of interlayer exciton and intralayer ligand-based excimer at 480 and 560 nm, respectively.

4) Furthermore, the authors should conduct light-polarization-dependent PL measurements to provide direct evidence for intra-layer and inter-layer exciton emission, see the details in Ref. [35].

Response: Many thanks for the reviewer's good suggestion. Light-polarization-dependent PL spectra of Ca-MOF crystal has been measured and shown in Fig. G. The direction of light polarization was adjusted by rotating the half-wave plate, while the sample was fixed on the quartz glass. When excited at 405 nm, the interlayer emission intensity increased steadily by varying the polarization angle from 40 to 190°. When excited at 460 nm, the intralayer emission intensity increased steadily by varying the polarization angle from 40 to 160°, following by a reduction at 190°. These two different light-polarization-dependent PL provide direct evidence for the intra-layer and inter-layer emissions, respectively.

Fig. G Light-polarization-dependent PL spectra of Ca-MOF single crystal with different polarization angle excited at 405 (a) and 460 nm (b).

5) Referring to the discussion section, the authors mentioned that “... dual-channel emission ... prompts the conceptual applications of ... optical memory.” The concept of “optical memory” is referred as to a material which exhibits a different optical property after being exposed to laser excitation. Here, the authors did not explain how their MOFs can be switched to have different emission properties after being exposed to a laser beam.

Response: We have tested the emission spectra of explored materials being exposed to a laser beam as shown in Fig. H. It indicated that the spectra were almost the same with comparison to those being excited by a Xe lamp light, though showing difference in the emission intensity. Therefore, we propose the “optical memory” mentioned in the manuscript should be rational, and continued work will be followed along this line in the future.

Fig. H Emission spectra of the explored materials being exposed to a laser beam of 405 nm.

To Reviewer #3

Comments:

In this paper the authors report a new Ca-MOF made using a fluorescent ligand, and which is because of this ligand fluorescent itself. This fluorescent MOF emits at two different wavelengths and the intensity of one of these emissions can be changed by sonicating the MOF in different solvents producing new nanoscale morphologies. The ability to tune the optical properties of a material through the formation of nanoscale phases is very topical, with many examples in purely inorganic materials (e.g. graphene, MoS₂), but this phenomenon has been rarely explored in MOFs. CaMOFs are relatively unusual, and the apparent stability of this material is itself quite exciting. The analysis of the data is good throughout, at the level which it has been carried out. The results in this paper are very intriguing but sometimes there could be more thorough interpretation and explanation with less speculation on properties. The two questions which seem unclear to me are:

What is the nano belt phase? It is described as a delaminated version of the Ca-MOF, but the powder pattern unambiguously proves it is not closely related to the Ca-MOF in terms of structure. There may not be free Ca ions, but this says nothing about the structure of the material. All explanations of the fluorescence of this material on the basis of particular structural changes are therefore unconfirmed. The authors do hint at the different structure of this material, but I think an upfront description of the change is necessary: to produce nanosheets of the Ca-MOF material through ultrasonication is a two step process and the

authors should be clear about this in text and figures. It would be interesting if the authors could suggest a structure for this phase, perhaps backed up through either surface sensitive diffraction measurements or high resolution microscopy.

Response: Many thanks for the reviewer's suggestion. As responded to the question of Reviewer #1, we have tried to apply for an EXAFS test for a detailed structure detection of the nano-phase materials. But unfortunately, due to the scarce availability and long queue, no chances can be obtained in the near future.

And as stated above, we tried another method, by soaking the single crystals of the Ca-MOF (LIFM-41) in methanol solvent containing little amount of water (2%, volume ratio) for 24 hours, water-substituted crystals Ca-MOF-H₂O were obtained. In the structure of Ca-MOF-H₂O, we can see that the originally coordinated DMF molecules are substituted by H₂O molecules, while the **metal-ligand layer is** basically intact. This is in support of our assumption, that the exfoliation and structural transformation of the Ca-MOF into different morphologies (including the nanobelt) can be induced by replacement of the coordinated DMF molecules, as detected by the EA and TG data summarized in **Table S3**. It is also noted that the transformation of Ca-MOF between different morphologies is **facilely reversible (Figure 4)**. This also proves that the coordination environment around the Ca(II) centers during morphological transformation processes should not be too dramatic, and the assumption of DMF substitution is plausible. And as suggested, **two step process might happened during the ultrasonication, replacement of DMF by water, and then breakdown of the layer structures**. The corresponding discussion and structure of Ca-MOF-H₂O have been added in the text and SI.

The description of the two kinds of emission could be clarified and the origin of the morphology-dependence is a little unclear. Solution measurements show that on reduced aggregation the lower energy band disappears, but in the solid state it is this lower energy band that remains on reduction of dimensionality. This is not the usual behaviour and the contrast between the solution and solid state data is not fully explained. The assignment of the 'interlayer' vs 'intralayer' bands seems to be somewhat circular (the band that disappears on delamination must be interlayer; the interlayer band disappears because of delamination) which means the interpretation of the physical mechanism for the observed changes is not justified. Why should a band that appears for ligands dissolved at low concentrations be assigned to interlayer emission in the solid? If this could be more clearly explained it would be very useful. A more complete explanation of these critical data would greatly benefit this paper.

Response: As typically accepted for organic chromophores, the band that appears at high energy for pure ligands dissolved in low concentration is assigned as interligand charge transfer (ILCT) emission, and the one at lower energy is assigned as the excimer emission, which appear in high concentration solution of the ligand by forming coalescence states and also in the solid state of Ca-MOF (**Figure S4**). While the high energy emission of Ca-MOF appeared at around 480 nm is ascribed to interlayer exciton emission, which is subject to the excitation wavelengths, and disappears upon delamination.

Other issues:

The spectroscopic fitting of the datasets should be able to yield quantitative information, but I can't seem to find this information reported. It would be very useful to understand the

datasets: particularly as some figures (e.g. Fig 2b) are extremely difficult to interpret. For Fig 2b in particular, perhaps a 2D data plot would be better? Or at least a consistent coloring of the line corresponding to the incident radiation (e.g. blue for low wavelength to red for high wavelength).

Response: Many thanks for the reviewer's good suggestion. In order to show the corresponding emitting colors of main peak, the line colors in Figure 2b was changed from purple to orange, basically in accordance with their emitting color as shown in CIE coordinates .

Fig. 1 Excitation-dependent emission of 2D Ca-MOF bulk crystals (inset shows the shifting tendency of CIE coordinates with different excitation wavelengths)

The proposed application is not very novel and seems somewhat unsuitable: the transformations described by the authors require ultrasonication and large scale morphological changes. This would be an extremely unwieldy memory device. This material could be useful for sensing or similar applications, but the proposed mechanism seems to be a long way from a realisable memory device.

Response: Many thanks for the reviewers suggestion. We have appended the emission spectra of the explored materials excited by a laser beam as shown in Fig. H, which gave basically the similar results with those being excited by a Xe lamp light. Therefore, “optical memory” mentioned in the manuscript in more practical manners might be rational by using laser as a resource. Of course, the present module is just an initialized hypothesis, and continued work will be followed along this line toward real applications in our laboratory.

Fig. H Emission spectra of the explored materials being exposed to a laser beam of 405 nm.

More minor issues:

Sometimes the choice of words is not quite appropriate. ‘scissoring’ is not really a physical description for the delamination of these materials. ‘peak splitting’ should be replaced by peak deconvolution or similar.

Response: ‘scissoring’ has been revised to ‘delaminating’, and ‘peak splitting’ has been revised to ‘peak deconvolution’ in the text.

Please report crystal size and absorption correction method in the CIF, and refinement strategy briefly in the SI.

Response: Crystal size and absorption correction method have been added into CIF. Refinement strategy has been added in SI.

In conclusion, this is a very interesting material, which undergoes a number of unusual chemical transformations with interesting optical properties. However, as it stands, the paper does not fully explain the chemical behaviour or the optical properties, which reduces its value significantly. Were this to be done, I would recommend this paper for publication.

Response: Many thanks for the reviewer’s positive evaluation, and we sincerely hope that the present version after revision according to the reviewers’ valuable suggestions can be accepted for publication on Nat. Commun.

Reviewers' comments:

Reviewer #1 (Remarks to the Author):

I would like to thank the authors for the reply. I believe a proper structural characterization of the nano-phases is still missing which is a crucial for understanding the properties and potential of this Ca-MOF. Additional structural measurements presented by the authors were performed on bulk crystals which may or may not be relevant to the behavior in the nano-phases. I hope the authors agree that it is better to wait a little bit longer and obtain the required structural data rather than speculating or assuming about the structure in the nano-scale.

Reviewer #2 (Remarks to the Author):

The authors have addressed my questions. Hence, I would like to recommend the publication of the revised manuscript.

Reviewer #3 (Remarks to the Author):

The authors have carried out a series of additional, useful experiments, and made some changes that improve the clarity of the text. However, the key scientific questions raised by myself and the other referees are not fully tackled.

First - the authors do not substantively address the fact that the nano belt phase has a powder diffraction pattern that proves that it is not a simple delaminated form of either their main phase or the new hydrate they report. While I understand the difficulties of obtaining synchrotron time at short notice, if the authors are unable to make any measurements to determine the structure of their material, it is incumbent upon them to be honest with the reader and not make unsupported claims about the structure. While it is no doubt plausible that a process as they outline could occur, and the additional experiments support the feasibility of the claim, the powder diffraction data contradict their proposal: the nano belt phase is not what the authors claim. In addition, the process for checking the possibility of free ligand being produced in sonication by solution NMR doesn't seem to make sense: the authors sonicate the ligand in D₂O, then wash with d₆-DMSO, but don't deal with the MOF? Is this a mistake in the text, or have I misunderstood the procedure?

Second - as highlighted particularly by referee 2, the description of the different optoelectronic properties is still very unclear, despite the additional measurements. I asked for a clarification of why the two bands should be assigned as interlayer and intralayer, and the authors in their response have only explained again the solution properties. Even if the relationship between the optical properties and measurements and the assignments to structural motifs are obvious to the authors, they need to make their argument explicit. This is particularly important considering the challenges associated with proving that a particular absorption is excitonic in nature (c.f. the follow-ups to Ref [35], *Advanced Materials*, 2017, 29, 47, 1702463 and *Advanced Materials*, 2017, 29, 47, 1705261) and that the authors make this the key claim of the paper.

Third - I accept that the authors would need to undertake substantial work to produce a working optical device, however, the authors should at least attempt to address the feasibility of their memory strategy: are the authors suggesting that the memory switching process would be as described in the paper (i.e. sonicating their material in solvents)? If so, the authors should at least highlight the obvious impracticalities. If not, the authors should explain how this device would in fact function.

On these three issues, I think it is necessary for the authors to either provide the data and

arguments to substantiate the claims made, or to acknowledge and be open the uncertainty with the reader. Until these are tackled I am unable to recommend this paper for publication, despite the interesting results contained within.

Response to Reviewers' comments:

Reviewer #1 (Remarks to the Author):

I would like to thank the authors for the reply. I believe a proper structural characterization of the nano-phases is still missing which is a crucial for understanding the properties and potential of this Ca-MOF. Additional structural measurements presented by the authors were performed on bulk crystals which may or may not be relevant to the behavior in the nano-phases. I hope the authors agree that it is better to wait a little bit longer and obtain the required structural data rather than speculating or assuming about the structure in the nano-scale.

A: Many thanks for the reviewer's suggestion. By utilization of Shanghai Synchrotron Radiation Facility (SSRF), we have now managed to conduct PDF tests for the nanobelt sample to be compared with that of the bulk crystal. As shown below, for the local coordination environment around Ca(II), the peaks at about 2.4 Å are similar for both the nanobelt and bulk crystal samples, relating to the Ca-O coordination bonds of Ca(II) with the ligands and DMF or water molecules, manifesting that the basic interplane metal-organic coordination structures are maintained in the nanobelt. Furthermore, at around 4 Å, another peak appears for the nanobelt, which is absent for the bulk crystal. According to the single crystal data obtained for Ca-MOF-H₂O as discussed below, there exist uncoordinated H₂O molecules in the crystal lattice of the water-changed samples, and the Ca-O distance from Ca(II) center to the nearest lattice H₂O molecule is around 4 Å, supporting the appearance of the 4 Å peak in the PDF test we obtained here. This further proves the substitution of DMF by water in the nanobelt sample, and is consistent with the TG and EA results as well.

Supplementary Figure 12 | PDF patterns of the nanobelt (green) and bulk crystal (blue) samples of Ca-MOF.

Reviewer #2 (Remarks to the Author):

The authors have addressed my questions. Hence, I would like to recommend the publication of the revised manuscript.

A: Many thanks for the reviewer's positive comments.

Reviewer #3 (Remarks to the Author):

The authors have carried out a series of additional, useful experiments, and made some changes that improve the clarity of the text. However, the key scientific questions raised by myself and the other referees are not fully tackled.

First - the authors do not substantively address the fact that the nano belt phase has a powder diffraction pattern that proves that it is not a simple delaminated form of either their main phase or the new hydrate they report. While I understand the difficulties of obtaining synchrotron time at short notice, if the authors are unable to make any measurements to determine the structure of their material, it is incumbent upon them to be honest with the reader and not make unsupported claims about the structure. While it is no doubt plausible that a process as they outline could occur, and the additional experiments support the feasibility of the claim, the powder diffraction data contradict their proposal: the nano belt phase is not what the authors claim. In addition, the process for checking the possibility of free ligand being produced in sonication by solution NMR doesn't seem to make sense: the authors sonicate the ligand in D₂O, then wash with d₆-DMSO, but don't deal with the MOF? Is this a mistake in the text, or have I misunderstood the procedure?

A: Many thanks for the reviewer's helpful question. As answered to reviewer#1, by utilization of Shanghai Synchrotron Radiation Facility (SSRF), we have now managed to conduct PDF tests for the nanobelt sample to be compared with that of the bulk crystal. As for the local coordination environment around Ca(II), the peaks at about 2.4 Å are similar for both the nanobelt and bulk crystal samples, relating to the Ca-O coordination bonds of Ca(II) with the ligands and DMF or water molecules, manifesting that the basic interplane metal-organic coordination structures are maintained in the nanobelt. Furthermore, at around 4 Å, another peak appears for the nanobelt, which is absent for the bulk crystal. According to the single crystal data obtained for Ca-MOF-H₂O as discussed below, there exist uncoordinated H₂O molecules in the crystal lattice of the water-changed samples, and the Ca-O distance from Ca(II) center to the nearest lattice H₂O molecule is around 4 Å, supporting the appearance of the 4 Å peak in the PDF test we obtained here. This further proves the substitution of DMF by water in the nanobelt sample, and is consistent with the TG and EA results as well. For the PXRD peak appeared at small angle below 5 degree for the nanobelt sample, this might be due to the formation of a larger pseudo-crystal lattice. But as confirmed

by the PDF results, the basic coordination structure in the nanobelts is in consistence with what we speculated before.

As for the NMR tests, we are very sorry for the mistake in the text of the R1 version, it should be “sonicate the MOF sample in D₂O, then wash with d₆-DMSO”.

Second - as highlighted particularly by referee 2, the description of the different optoelectronic properties is still very unclear, despite the additional measurements. I asked for a clarification of why the two bands should be assigned as interlayer and intralayer, and the authors in their response have only explained again the solution properties. Even if the relationship between the optical properties and measurements and the assignments to structural motifs are obvious to the authors, they need to make their argument explicit. This is particularly important considering the challenges associated with proving that a particular absorption is excitonic in nature (c.f. the follow-ups to Ref [35], *Advanced Materials*, 2017, 29, 47, 1702463 and *Advanced Materials*, 2017, 29, 47, 1705261) and that the authors make this the key claim of the paper.

Response: Many thanks for the reviewer’s suggestion. As an alteration to the method in the literature to directly measure the absorption as a function of radiation polarization and power, we measured light-polarization-dependent PL spectra of Ca-MOF crystal due to limitation of instrument for absorption test under polarized radiation. And as supplied in Supplementary Figure 7 in the previously revised version as a response to Reviewer #2, the direction of light polarization was adjusted by rotating the half-wave plate, while the sample was fixed on the quartz glass. When excited at 405 nm, the interlayer emission intensity increased steadily by varying the polarization angle from 40 to 190°. When excited at 460 nm, the intralayer emission intensity increased steadily by varying the polarization angle from 40 to 160°, following by a reduction at 190°. These two different light-polarization-dependent PL provide evidence for the intra-layer and inter-layer emissions, respectively. And we also sense that more direct evidence might still be needed, so we add the statement to the readers about the follow-up studies needed and leave uncertainty open.

Supplementary Figure 7 | Light-polarization-dependent PL spectra of Ca-MOF single crystal with different polarization angle excited at 405 (a) and 460 nm (b).

Third - I accept that the authors would need to undertake substantial work

to produce a working optical device, however, the authors should at least attempt to address the feasibility of their memory strategy: are the authors suggesting that the memory switching process would be as described in the paper (i.e. sonicating their material in solvents)? If so, the authors should at least highlight the obvious impracticalities. If not, the authors should explain how this device would in fact function.

A: Many thanks for the reviewer's helpful suggestion. In the present stage, it is indeed of great impracticalities to make a real memory device based on our existing conditions and techniques. Although the working theory might be relied on the switching of different states of the samples using such methods of sonicating in solvents. But this method is indeed difficult to be accomplished for a formed device. Otherwisely, grinding or rubbing under humid conditions might also be able to achieve the purpose. But since in the present conditions, we can not fullfill to produce a working device, therefore, as suggested by the reviewer, we add uncertainty and impracticality statement in the text, and do hope that the publication of this work can stimulate further and deeper study in this field, and achieve the final goal of device fabrication and operation.

On these three issues, I think it is necessary for the authors to either provide the data and arguments to substantiate the claims made, or to acknowledge and be open the uncertainty with the reader. Until these are tackled I am unable to recommend this paper for publication, despite the interesting results contained within.

REVIEWERS' COMMENTS:

Reviewer #1 (Remarks to the Author):

I find the additional experiments satisfying. Manuscript requires some minor revision to integrate the PDF measurements better. For example, PDF acronym is not explained and some brief statement about the technique is required. Please report resolution and software/methods used for collecting and reducing the PDF data.

Reviewer #3 (Remarks to the Author):

The authors have now presented additional evidence for their claims, and reduced the certainty with which they state them. I am satisfied that the paper as is now supported by the data presented. I therefore am in general happy to recommend publication, after some minor changes.

The authors have added some pair distribution function data, but include limited information about the data collection, and none about how the data were processed (e.g. GudrunX, PDFGetX etc.), and only include an abbreviated fraction of the data. For future readers it would be very useful to include this information (which ought to be straightforward for the reviewers).